# Lagrangian Partition Functions Subject to a Fixed Spatial Volume Constraint in the Lovelock Theory

**DOI:** 10.3390/e26040291

**Published:** 2024-03-27

**Authors:** Mengqi Lu, Robert B. Mann

**Affiliations:** 1Department of Physics and Astronomy, University of Waterloo, Waterloo, ON N2L 3G1, Canada; mengqi.lu@uwaterloo.ca; 2Perimeter Institute for Theoretical Physics, 31 Caroline St. N., Waterloo, ON N2L 2Y5, Canada

**Keywords:** de Sitter, gravitational Hilbert space, Lovelock gravity, cosmological entropy

## Abstract

We evaluate here the quantum gravity partition function that counts the dimension of the Hilbert space of a simply connected spatial region of a fixed proper volume in the context of Lovelock gravity, generalizing the results for Einstein gravity. It is found that there are sphere saddle metrics for a partition function at a fixed spatial volume in Lovelock theory. Those stationary points take exactly the same forms as in Einstein gravity. The logarithm of *Z* corresponding to a zero effective cosmological constant indicates that the Bekenstein–Hawking entropy of the boundary area and that corresponding to a positive effective cosmological constant points to the Wald entropy of the boundary area. We also show the existence of zeroth-order phase transitions between different vacua, a phenomenon distinct from Einstein gravity.

## 1. Introduction

The Euclidean action approach to the gravitational partition function *Z* was originally formulated by Gibbons and Hawking [1]. They evaluated the path integral over spacetime geometries by the saddle point approximation and found that the logarithm of *Z* for a de Sitter (dS) spacetime is a quarter of the cosmological horizon area, indicating that (despite the absence of an event horizon) the corresponding thermodynamic potential was the dS entropy, analogous to the Bekenstein–Hawking entropy of a black hole [2,3]. This justified the general holographic nature of gravitational entropy, the concept of which is not limited to a black hole horizon.

However, interpreting these results involves addressing nuanced inquiries. In the context of statistical mechanics, the thermodynamic equilibrium of a system is determined by extremizing a thermodynamic potential. Without any constraint, the concept of thermodynamic equilibrium loses its generality since we can hardly guarantee an unconstrained system to have extrema in its thermodynamic potentials. This gives rise to the question as to what constraint other than fixing the temperature is imposed implicitly in specifying the canonical ensemble for the dS vacuum. Furthermore, the interpretation of the dS entropy remains ambiguous [4]. For a Euclidean dS space, the quasi-local energy vanishes identically due to the absence of a boundary. Given the vanishing Hamiltonian, Fischler [5] and Banks [6] further argued that the finite de Sitter entropy equates to the logarithm of the dimension of the Hilbert space since the partition function reduces to the trace of the identity operator. This assertion has since received several lines of support [7,8,9,10,11,12,13,14]. Furthermore, the concept of gravitational entropy requires a general description since it can be associated with the area of any boundary separating a region of space [15,16], and not only to the area of black hole or dS horizons.

Recently, Jacobson and Visser [17] considered some of these issues by adding a spatial volume constraint to the Euclidean path integral. They proposed a partition function with a Lagrange multiplier λ(τ) of the form
(1)$$Z=\;\int \int \;D\lambda Dg\exp(-I_{E}+\int d\tau \lambda \left(\tau \right)\hat{C}),$$
in *D* spacetime dimensions, where the explicit expression $\hat{C}$ is implemented by
(2)$$\hat{C}=\int d^{D-1}x\sqrt{\gamma }-V,$$
and the action $I_{E}$ is the Euclidean Einstein–Hilbert term plus a bare cosmological constant $\Lambda_{0}$,
(3)$$I_{E}=-\frac{1}{16\pi }\int_{M_{E}}d\tau d^{D-1}x\sqrt{g}\left(R-2\Lambda_{0}\right).$$
The first term in the exponential in (Equation 1) is performed on a Euclidean manifold $M_{E}$ of dimension *D* with a periodic boundary condition on the imaginary time $x^{0}=\tau$. The second term imposes the volume constraint on each constant-τ slice with induced metric $\gamma_{ij}$. The temperature, as the reciprocal of the period β in τ, arises by eliminating the conical singularity at the horizon. Thus, the fixed temperature, or the size of $\partial M_{E}$ in the τ direction, and the fixed volume together define a canonical ensemble. The partition function (Equation 1) should be considered a provisional prescription for manifolds $M_{E}$ with closed spatial sections; otherwise, a Hawking–Gibbons–York (HGY) term and counterterms should be included. Together with the closedness in the τ direction, the desired saddle manifold should have a closed topology similar to the original Euclidean dS space.

The partition function (Equation 1) was shown [17] to admit spherical saddle metrics of the form
(4)$$ds_{E}^{2}=h\left(r\right)d\tau^{2}+\frac{1}{f(r)}dr^{2}+r^{2}d\Omega_{D-2}^{2}$$
for both zero and nonzero $\Lambda_{0}$. Here, $d\Omega_{D-1}^{2}$ is the induced metric on a D−2 dimensional sphere. Two metric components $g_{\tau \tau }=h$ and $g_{rr}=1/f$ are some radial dependent functions that solve the field equations, with *h* chosen to satisfy the boundary condition
(5)$$\frac{d\sqrt{h}}{dl}{|}_{l=0}=2\pi T=\frac{2\pi }{\beta }$$
where *l* is the radial proper distance from the Euclidean horizon. When such a stationary point dominates, the partition function can be approximated by the minimum saddle action $I_{E}^{\min}$, thus yielding $Z\approx \exp\left(-I_{E}^{\min}\right)$ due to the vanishing of $\hat{C}$ for any saddles. The logarithm of *Z* then turns out to be the Bekenstein–Hawking entropy, i.e., one quarter of the horizon area of the boundary. Hence, the dS partition function can be understood as a special case of (Equation 1) where the boundary of *V* matches that of the dS horizon. Furthermore, this result not only justifies the interpretation of the dS entropy as the measure of the dimension of Hilbert space, but also generalizes the description of entropy (originally associated with an area) to a volume separated by a boundary.

Motivated by this approach, we investigate here how these conclusions are modified in higher curvature theories of gravity. In the low energy effective action of whatever theory provides the UV completion to General Relativity, such theories are generically expected to appear, and are known to modify the entropy of a black hole [18]. It is therefore natural to inquire as to how they affect the entropy of a volume of space. For simplicity, we consider Lovelock gravity since the quasi-local energy in Lovelock theory is given by an integral on the system’s boundary [19], affording the same interpretation of dS entropy as the dimension of Hilbert space as for GR [5,6].

## 2. Towards Lovelock Theory

The only change we make to (Equation 1) is the replacement of $I_{E}$ by the general Lovelock action [20],
(6)$$I_{E}=\frac{1}{16\pi }\sum_{p=0}\int_{M_{E}}\frac{\alpha_{p}}{(D-2p)!}R^{\left(p\right)}$$
where the $\alpha_{p}$ are coupling constants. The *D*-form $R^{\left(p\right)}$ is given by the antisymmetrization of exterior products of *p* curvature 2-forms $R^{ab}$, which is antisymmetric in the two indices, with D−2p vielbeins $e^{a}$. It reads
(7)$$R^{\left(p\right)}=\epsilon_{a_{1}\cdots a_{D}}R^{a_{1}a_{2}\cdots a_{2p-1}a_{2p}}\wedge e^{a_{2p+1}\cdots a_{D}},$$
where the total antisymmetric tensor ϵ is defined by $\epsilon_{0\cdots (D-1)}=+1$ in an orthonormal basis. For convenience, we write
$$R^{a_{1}a_{2}\cdots a_{2n-1}a_{2n}}:=R^{a_{1}a_{2}}\wedge \cdots \wedge R^{a_{2n-1}a_{2n}}$$
$$e^{a_{1}a_{2}\cdots a_{n}}:=e^{a_{1}}\wedge \cdots \wedge e^{a_{n}}.$$
The zeroth and the first order couplings are set to $2\Lambda_{0}$ and −1, respectively, for consistency with (Equation 3), rendering (Equation 6) a higher curvature modification of GR.

The Lovelock action can be easily evaluated for the ansatz (Equation 4) in the language of differential forms. In the vielbein frame defined by
$$e^{0}=\sqrt{h}dt,\;e^{1}=\frac{1}{\sqrt{f}}dr,\;e^{2}=rdx^{2},\;e^{i}=r\prod_{\xi =2}^{i-1}\sin x^{\xi }dx^{i}\;\left(i\geq 3\right),$$
the curvature 2-forms can be directly obtained by manipulating those vielbeins with Cartan formulae for zero-torsion and metric compatible geometries. The non-vanishing and non-repeating components for $R^{ab}$ are then
(8)$$\begin{array}{c} R^{01}=-\sqrt{\frac{f}{h}}{\left(\sqrt{\frac{f}{h}}\frac{h^{\prime }}{2}\right)}^{\prime }e^{0}\wedge e^{1},\;R^{0i}=-\frac{fh^{\prime }}{2rh}e^{0}\wedge e^{i}, \end{array}$$
$$R^{1i}=-\frac{f^{\prime }}{2r}e^{1}\wedge e^{i},\;R^{ij}=\frac{1-f}{r^{2}}e^{i}\wedge e^{j}$$
where ′ denotes the derivative with respect to *r*, numbers are used for all coordinate directions, and the Latin indices i,j represent coordinate indices on the sphere $S^{D-2}$.

The Lovelock density (Equation 7) of order *p* can be rewritten as
(9)$$R^{\left(p\right)}=\sum_{\sigma \in S_{D}}\mathrm{sgn}\left(\sigma \right)R^{\sigma (0)\sigma (1)\cdots \sigma (2p-2)\sigma (2p-1)}\wedge e^{\sigma (2p)\cdots \sigma (D-1)},$$
where the Levi–Civita tensor is replaced by permutations of the cyclic group $S_{D}$ of order *D*. Since the curvature forms $R^{ab}$ are proportional to $e^{a}\wedge e^{b}$ and each index appears exactly once in each permutation, we conclude all indices are distinct in the $R^{\cdots }$ term in (Equation 9). This means that there are only five possible pairings of the terms in (Equation 8), yielding
$$R^{\left(p\right)}{|}_{h,f}={\left({\tilde{R}}^{SS}\right)}^{p-2}\left[A{\tilde{R}}^{01}{\tilde{R}}^{SS}+B{\tilde{R}}^{0S}{\tilde{R}}^{1S}+C{\tilde{R}}^{0S}{\tilde{R}}^{SS}+D{\tilde{R}}^{1S}{\tilde{R}}^{SS}+E{\left({\tilde{R}}^{SS}\right)}^{2}\right]e^{0\cdots D-1},$$
for the on-shell density, where
$${\tilde{R}}^{01}=-\sqrt{\frac{f}{h}}{\left(\sqrt{\frac{f}{h}}\frac{h^{\prime }}{2}\right)}^{\prime },\;{\tilde{R}}^{0S}=-\frac{fh^{\prime }}{2rh},\;{\tilde{R}}^{1S}=-\frac{f^{\prime }}{2r},\;{\tilde{R}}^{SS}=\frac{1-f}{r^{2}}$$
are the four possible coefficients of the curvature forms. The quantities *A* to *E* are
A=2p1(D−2)!D=C=(2p)!D−22p−1(D−2p)!B=8p2(D−2)!E=(2p)!D−22p(D−2p)!
and are coefficients counting the duplicates of each building block. For instance, *A* represents the number of permutations of {0,1} that appear in a single $R^{ab}$—hence, a factor of 2 shows up for the two different arrangements of {0,1}; the second factor counts all different places in which $R^{01}$ appears; the remaining (D−2)! term corresponds to all the permutations with 0,1 fixed. Other parameters can be obtained in a similar way. Collecting all building blocks together, we finally arrive at
(10)$$\begin{array}{cc} R^{\left(p\right)}{|}_{h,f}= & \sqrt{\left|g\right|}d^{D}x\left(D-2\right)!{\left(\frac{1-f}{r^{2}}\right)}^{p-2}\{-p\sqrt{\frac{f}{h}}{\left(\sqrt{\frac{f}{h}}h^{\prime }\right)}^{\prime }\left(\frac{1-f}{r^{2}}\right)\\ & \;+p\left(p-1\right)\left(\frac{ff^{\prime }h^{\prime }}{hr^{2}}\right)-\left(D-2p\right)p\left(\frac{1-f}{r^{3}}\right)\left(\frac{fh^{\prime }}{h}+f^{\prime }\right)\\ & \;\;\;\;\;+\left(D-2p\right)\left(D-2p-1\right){\left(\frac{1-f}{r^{2}}\right)}^{2}\}. \end{array}$$

The field equations for the metric ansatz (Equation 4) are obtained through extremizing the volume-fixed action with respect to $g_{00}=h\left(r\right)$ and $g^{11}=f\left(r\right)$ simultaneously. Given the tensor defined as $E^{ab}:=\frac{1}{\sqrt{-g}}\delta I_{E}\left[g\right]/\delta g_{ab}$, the relations (Equation 6) and (Equation 10), together with the volume constraint in (Equation 1) give rise to
$$E^{00}=\frac{1}{\sqrt{\left|g\right|}}\frac{\delta I_{E}\left[h,f\right]}{\delta h}=0,\;E_{11}=\frac{1}{\sqrt{\left|g\right|}}\frac{\delta I_{E}\left[h,f\right]}{\delta f}=\frac{1}{\sqrt{\left|g\right|}}\frac{\lambda }{2}f^{-\frac{3}{2}}r^{D-2},$$
which can be equivalently cast into the form
$$E_{\;0}^{0}=0,\;E_{\;0}^{0}+E_{\;1}^{1}=\frac{\lambda }{2\sqrt{h}}.$$
They respectively correspond to the explicit expansions
(11)$$0=\sum_{p=0}^{p_{\max}}\alpha_{p}\frac{(D-2)!}{2(D-2p-1)!}\frac{{\left[r^{D-2p-1}{\left(1-f\right)}^{p}\right]}^{\prime }}{r^{D-2}},$$
(12)$$\frac{8\pi \lambda }{\sqrt{h}}=\sum_{p=0}^{p_{\max}}\alpha_{p}\frac{(D-2)!}{2(D-2p-1)!}pr^{1-2p}{\left(1-f\right)}^{p-1}h{\left(\frac{f}{h}\right)}^{\prime },$$
which are consistent with the result in [21] but with some sign flips due to the positive definiteness of the Euclideanized metric (Equation 4).

The metric function *f* is fully characterized by the first equation, and so is the same as the one for unconstrained dS vacua since the volume constraint makes no contribution here. The spatial metric component thus takes the specific form given by
(13)$$f\left(r\right)=1-\Lambda r^{2}$$
with an effective cosmological constant Λ>0 that characterizes the de Sitter horizon by $1/\sqrt{\Lambda }$, the value of which satisfies the algebraic equation
(14)H(Λ)=0,
where H(x) is defined as a polynomial of the form
(15)$$H\left(x\right):=\sum_{p=0}^{p_{\max}}\alpha_{p}\frac{(D-2)!}{(D-2p-1)!}x^{p}.$$
where we have taken out a factor of (D−1) to simplify subsequent expressions.

The non-trivial effects of the volume constraint take place in the second equation, which causes *h* to deviate from its value of unity for unconstrained dS vacua. More precisely, the explicit relation between λ and the metric component *h* is given by the direct substitution of (Equation 13) to (Equation 12), which yields
(16)$$\frac{16\pi \lambda }{\sqrt{h}}=\frac{h}{r}{\left(\frac{f}{h}\right)}^{\prime }H^{\prime }\left(\Lambda \right),$$
where $H^{\prime }\left(x\right)$ is the first derivative of *H* with respect to its argument, and λ should be a pure constant since no factor is τ-dependent in the equation.

Note that (Equation 16) imposes the same structure on *h* as in GR [17], which implies that the solutions for all Lovelock theories (GR included) are qualitatively the same. The only distinction between the various theories is that (Equation 14) admits multiple solutions that depend on the coupling constants $\alpha_{p}$. These correspond to various distinct Lovelock vacua that are characterized by different effective cosmological constants. These constants should be limited to non-negative values since only those that have closed topologies are consistent with a vanishing Gibbons–Hawking term in the proposed action [17]. The allowed saddles can be classified into two categories, Λ=0 and Λ>0, and each contributes to the partition function qualitatively differently. In other words, the category of $I_{E}^{\min}$ should be clearly specified when approximating the density of states by saddle points. In subsequent sections, we will explore generic solutions for h(r) and evaluate *Z* as it is maximized by each type.

## 3. $I_{E}^{\min}$ for a Vanishing Cosmological Constant

Whenever the saddle of vanishing Λ minimizes the action, the value of $I_{E}^{\min}$ can be obtained by substituting the solution of (Equation 16) into (Equation 10) under the assumption Λ=0. In this scenario, *f* reduces to 1, and the solution for the ττ component of the metric becomes
(17)$$h\left(r\right)={\left(\frac{4\pi \lambda }{D-2}\right)}^{2}{\left(r_{v}^{2}-r^{2}\right)}^{2}$$
where the factor of D−2 comes from $H^{\prime }\left(0\right)$. The constant of integration $r_{v}$ satisfying $h(r_{v})=0$ locates the horizon at the boundary of the volume ball and is determined by the spatial volume constraint
(18)$$V=\Omega_{D-2}\int_{0}^{r_{v}}r^{D-2}dr.$$

Along with the metric, the range of integration of (Equation 6) needs to be clarified. The spatial range is limited to the volume *V*, whereas the Euclidean time τ is restricted to a closed loop with period β, whose value is determined from eliminating the conical singularity of the manifold $M_{E}$ at $r_{v}$. In the vicinity of the horizon, the metric is approximately
$$ds_{E}^{2}{|}_{l\to 0}\approx {\left(\frac{8\pi \lambda r_{v}}{D-2}\right)}^{2}l^{2}d\tau^{2}+dl^{2}+r_{v}^{2}d\Omega_{D-2}^{2}$$
where $r\approx r_{v}-l$. It is easy to see that the quantity $\phi :=8\pi r_{v}\lambda \tau /\left(D-2\right)$ must have a period of 2π, or equivalently τ has a period of $\beta =\left(D-2\right)/\left(4\lambda r_{v}\right)$, so that the conical singularity is removed. This establishes the relationship between β and λ; in terms of β, the metric reads
(19)$$ds_{E}^{2}=\frac{{\left(r_{v}^{2}-r^{2}\right)}^{2}}{4r_{v}^{2}}d{\left(\frac{2\pi \tau }{\beta }\right)}^{2}+dr^{2}+r^{2}d\Omega_{D-2}^{2}$$
and its accompanying Ricci scalar is
$$R=\frac{4(D-1)}{r_{v}^{2}-r^{2}}\;.$$
An explicit calculation then shows that the action is
$$\begin{array}{cc} I_{E}^{\min} & =-\frac{1}{16\pi }\int_{0}^{\beta }d\tau \int \sqrt{g}d^{D-1}xR\\ & =-\Omega_{D-2}\int_{0}^{2\pi }d\xi \int_{0}^{r_{v}}\frac{\left(D-1\right)r^{D-2}}{8\pi r_{v}}dr\\ & =-\frac{\Omega_{D-2}r_{v}^{D-2}}{4}=-\frac{A_{v}}{4} \end{array}$$
where
$$\xi :=2\pi \tau /\beta \;A_{v}:=\Omega_{D-2}r_{v}^{D-2}\;.$$
We see that the Ricci scalar is the only contributing term to the Lagrangian density, since (Equation 10) becomes trivial for any p≥2 when f=1. The cosmological term is necessarily zero because $H\left(0\right)=2\Lambda_{0}/\left(D-1\right)=0$. The final result clearly indicates that the Bekenstein–Hawking entropy, which is one quarter of the boundary area $A_{v}$, is the extremal value of the action in this case for all Lovelock theories.

## 4. $I_{E}^{\min}$ for a Positive Effective Cosmological Constant

In the case where the action is minimized by some solution with a positive Λ, the function *h* satisfying (Equation 16) takes the form
(20)$$h\left(r\right)={\left(\frac{8\pi \lambda }{\Lambda H^{\prime }}\right)}^{2}{\left(1-\frac{\sqrt{1-\Lambda r^{2}}}{\sqrt{1-\Lambda r_{v}^{2}}}\right)}^{2},$$
where Λ is some positive quantity that solves H=0. Again, the constant of integration $\sqrt{1-\Lambda r_{v}^{2}}$ implies the existence of a singularity at the horizon $r_{v}$. The multiplier λ can again be written in terms of the period β by eliminating the conical singularity, which yields
(21)$$\beta =\frac{H^{\prime }\sqrt{1-\Lambda r_{v}^{2}}}{4\lambda r_{v}}.$$
Thus, the saddle metric for a constrained dS vacuum reads
(22)$$ds_{E}^{2}=\frac{1}{\Lambda }\left[{\left(\frac{\cos\chi -\cos\chi_{v}}{\sin\chi_{v}}\right)}^{2}d{\left(\frac{2\pi \tau }{\beta }\right)}^{2}+d\chi^{2}+\sin^{2}\chi d\Omega_{D-2}^{2}\right]$$
where the reparametrization $\sin\chi =\sqrt{\Lambda }r$ is applied for simplification. If the horizon matches the cosmological one, or equivalently $\chi_{v}=\pi /2$, the metric (Equation 22) reduces to the static patch for unconstrained Euclidean dS spacetime, namely
$$ds_{\mathrm{EdS}}^{2}=\frac{1}{\Lambda }\left[\cos\chi^{2}d\tau^{2}+d\chi^{2}+d\Omega_{D-2}^{2}\right].$$

For the Lagrangian density (Equation 10), we obtain
(23)$$R^{\left(p\right)}=\left(D-1\right)!\Lambda^{p}\left[D+\frac{2p\cos\chi_{v}}{\cos\chi -\cos\chi_{v}}\right]\sqrt{g}d\tau \wedge d^{D-1}x,$$
for the saddle (Equation 22). The action then becomes
(24)$$\begin{array}{c} I_{E}^{\min}=\frac{A_{v}}{8}\sum_{p}\frac{D(D-2)!}{(D-2p)!}\alpha_{p}\Lambda^{p-1}+\frac{(D-1)V}{8}\frac{\cos\chi_{v}}{\sin\chi_{v}}\Lambda^{-1/2}H\left(\Lambda \right), \end{array}$$
where $\chi_{v}$ is constrained by
(25)$$V=\Lambda^{\frac{1-D}{2}}\Omega_{D-2}\int_{0}^{\chi_{v}}{\left(\sin\chi \right)}^{D-2}d\chi =\Omega_{D-2}\int_{0}^{r_{v}}\frac{r^{D-2}}{\sqrt{1-\Lambda r^{2}}}dr,$$
and the horizon area $A_{v}$ is given by
(26)$$A_{v}=\Lambda^{\frac{2-D}{2}}\Omega_{D-2}{\left(\sin\chi_{v}\right)}^{D-2}=\Omega_{D-2}r_{v}^{D-2}.$$
Note that the second term in the action vanishes since H(Λ)=0, so its value is independent of $\chi_{v}$. This means the action is the same as the unconstrained dS action but with $A_{v}$ replaced by the area of cosmological horizon. The result can be further simplified by setting H=0, for which we rewrite the factor *D* as D−2p+2p,
(27)$$I_{E}^{\min}=\frac{A_{v}}{8}\sum_{p}\frac{(D-2)!}{(D-2p-1)!}\alpha_{p}\Lambda^{p-1}+\frac{A_{v}}{4}\sum_{p}\frac{(D-2)!}{(D-2p)!}p\alpha_{p}\Lambda^{p-1}.$$
For non-zero Λ, the first term vanishes as it is proportional to H(Λ). The second one is the negative of the Wald entropy for the boundary area $A_{v}$, consistent with [22].

## 5. Phase Transitions between Different Vacua

As mentioned above, the approximation for *Z* only works for cases where saddle points exist. Its validity necessarily confines the system into a certain amount of volume. This value is bounded by the total spatial volume screened inside the cosmological horizon in GR where only one vacuum solution could exist. However, Lovelock gravities extend the possibilities for vacuum solutions, so we expect a discontinuity to appear in *S* when *V* reaches an intermediate value between two cosmological spatial volumes if the saddle point dominating for a smaller *V* no longer remains a solution due to the volume exceeding its maximum. For a dS vacuum, its free energy reduces to the gravitational entropy due to the lack of quasi-local energy
F(T,V)=−TlnZ=E−TS=−TS.
Therefore, a discontinuity in *S* implies a zeroth-order phase transition at a given temperature.

We take Gauss–Bonnet gravity in D≥5 to illustrate this idea. In this scenario, possible dS vacua are characterized by non-negative solutions of
(28)$$H\left(\Lambda \right)=\frac{2\Lambda_{0}}{D-1}-\frac{(D-2)!}{(D-3)!}\Lambda +\alpha_{2}\frac{(D-2)!}{(D-5)!}\Lambda^{2}=0$$
using (Equation 14). Consider the parameter space where (Equation 28) has two non-negative solutions. This forces the sum and the product of two solutions to satisfy
(29)$$\Lambda_{+}+\Lambda_{-}=\frac{1}{\alpha_{2}\left(D-3\right)\left(D-4\right)}\geq 0,\;\Lambda_{+}\Lambda_{-}=\frac{2\Lambda_{0}\left(D-5\right)!}{\alpha_{2}\left(D-1\right)!}\geq 0,$$
which further reduce to $\alpha_{2}>0$ and $\Lambda_{0}\geq 0$. Furthermore, (Equation 28) simplifies the saddle action to be
(30)$$I_{E}=\frac{A_{v}\left(V,\Lambda \right)}{4}\left[\frac{D}{D-4}-\Upsilon \right],\;\left\{\begin{array}{c} \Upsilon =\frac{2(D-2)}{\left(D-4\right)},\;\;\;\;\mathrm{if}\;\Lambda =0\\ \Upsilon =\frac{4\Lambda_{0}}{(D-1)(D-4)\Lambda },\;\mathrm{if}\;\Lambda \neq 0, \end{array}\right.$$
where Λ is a solution to (Equation 28), and $A_{v}$ is induced by (Equation 26) together with the constraint (Equation 25). An immediate conclusion we can draw from (Equation 25) is that, when *V* is sufficiently small such that both saddles exist, a smaller Λ implies a greater $r_{v}$ and thus a larger boundary area at a given volume. Note that a smaller Λ vacuum has a larger volume bound; in order to construct such a transition, we eliminate cases where the saddle with the smaller Λ minimizes the action for any volume smaller than the maximum *V* that corresponds to the other solution. In other words, to have transitions, larger Λ must have smaller action where this saddle exists.

It is easy to see $\Lambda_{0}=0$ implies one negative and one positive action for Λ=0 and Λ≠0, respectively. The solution with zero Λ always dominates the partition function and thus yields no interesting transition behaviour. For a positive $\Lambda_{0}$, we divide the discussion into three cases.

**D/(D−4)−Υ<0 for both vacua**. In this case, the larger Λ does not dominate the action for any *V*, since its $A_{v}$ is smaller and the square bracket in (Equation 30) is less negative. Thus, this case is trivial.**D/(D−4)−Υ has a different sign for each vacuum**. Here the solution with D/(D−4)−Υ>0 has a larger Λ and larger action. Thus, this case also produces a trivial effect.**D/(D−4)−Υ>0 for both vacua**. This case likely allows a discontinuity since $A_{v}$ is larger and D/(D−4)−Υ is less positive for the solution with a smaller Λ. However, we find that, at least for even dimensions, the smaller Λ solution always dominates the partition function as long as *V* is smaller than its cosmological volume. This is because, in order to ensure that (Equation 30) has the desired properties, we need not only two degrees of freedom from $\Lambda_{\pm }$ but also that from $\Lambda_{0}$.

In summary, a phase transition is generically not allowed in the Gauss–Bonnet theory. However, the situation differs in cubic (or higher-curvature) Lovelock gravity, since there are more coupling constants, but the number of solutions required to make the transition take place is still only two.

As an example, we find that zeroth-order transitions can occur in cubic Lovelock gravity theory in D≥7. Noting that F/T=−lnZ=I, as shown in Figure 1, we can see three different vacua emerge in a particular case of cubic Lovelock gravity where the vacuum with a larger Λ has the smaller action. Meanwhile, a larger Λ implies a smaller cosmological volume. Thus, two zeroth-order phase transitions appear in the free energy diagram at a fixed temperature, as shown in Figure 1.

Although our examples have been for even spatial dimensions, we expect similar results to hold for odd dimensions as well. In this case, the integral in volume (Equation 25) contains a part related to an inverse trigonometric function, and solving for $r_{v}$ for a given *V* must be carried out numerically.

## 6. Conclusions

Our results support the interpretation of the dS entropy as the dimension of the Hilbert space of all states constrained in a spatial volume, even when higher-curvature corrections to Einstein gravity are taken into account.

We have found that such corrections, as described by Lovelock gravity, generally yield metrics of constrained dS vacua that take the same form as that in Einstein gravity [17]. However, instead of the unique dS horizon radius given by the bare cosmological constant, we now have multiple vacua characterized by the non-negative zero points of (Equation 14). Classically, these different vacua correspond to spacetimes of differing constant curvature, with a given value of $\alpha_{0}$ in (Equation 6) in general up to $n=p_{\max}$ distinct possible solutions. Physically, only one of these solutions will be realized; there is no physical criterion for preferring one over another.

In the context of quantum gravity, the situation is different. As long as the partition function is dominated by one of these saddles, the logarithm of *Z* is given by the Wald entropy of the boundary. Hence, within a semiclassical static patch in any of the possible de Sitter spaces admitted by Lovelock gravity, the semiclassical, gravitationally dressed vacuum state of such a patch is close to a maximally mixed state. As in general relativity, the inclusion of matter fields will not modify our calculation provided they vanish in the saddle configuration. Consequently, Lovelock gravity theories are commensurate with the “maximal vacuum entanglement hypothesis”, namely that the semiclassical, gravitationally dressed vacuum state maximizes the entanglement entropy of gravity and matter in small regions at fixed volume. Quantum mechanically, once such a region no longer minimizes the Euclidean action due to volume expansion, a phase transition via quantum tunnelling to a vacuum of larger entropy (lower free energy) will occur, as shown in Figure 1.

Understanding the entropy of cosmological horizons is a necessary ingredient in unravelling the information paradox. It has recently been shown in the context of Einstein gravity that the dimension of the Hilbert space remains the exponential of the Gibbons–Hawking entropy, even when considering quantum corrections [14]. It would be interesting to discover what the effects of higher-curvature are on this relationship.

## Figures and Tables

**Figure 1 entropy-26-00291-f001:**
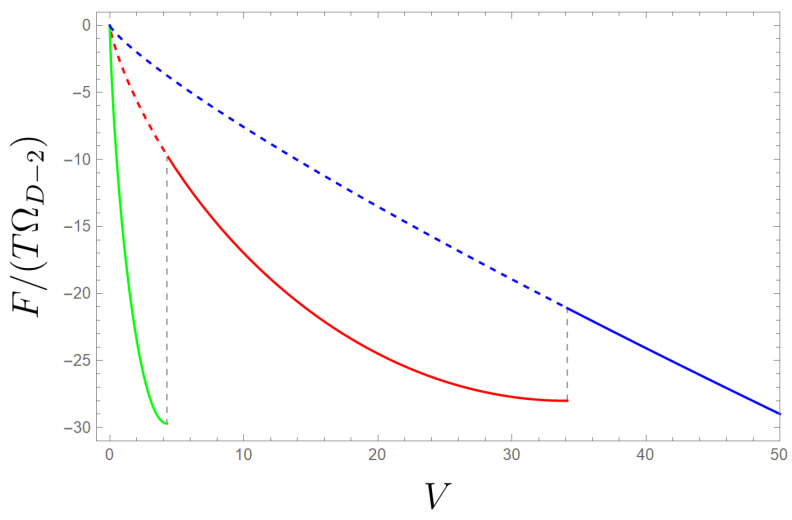
Zeroth-order phase transitions between three vacua of cubic Lovelock gravity with $\Lambda_{0}=0$, $\alpha_{2}=1/2,\alpha_{3}=1/3$ in D=7. Green corresponds to the vacuum with Λ=1/2; red corresponds to Λ=1/4; blue corresponds to Λ=0. The figure depicts the free energy as a function of *V* for the stable phases of the system, showing that, at large *V*, the Λ=0 vacuum (blue) dominates; at intermediate values of *V*, the Λ=1/4 vacuum (red) dominates; at small *V*, the Λ=1/2 vacuum (green) dominates.

## Data Availability

Data are contained within the article.

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
