# Peer review of "Lagrangian Partition Functions Subject to a Fixed Spatial Volume Constraint in the Lovelock Theory"

_entropy, 2024, doi:10.3390/e26040291_

Round 1
Reviewer 1 Report
Comments and Suggestions for Authors
I have examined this manuscript, in which the authors say tthey evaluate the quantum gravity partition function counting the dimension of the Hilbert space of a simply connected spatial region of fixed proper volume in the context of Lovelock gravity. I have not much to say on this short work, although it is not clear the significance of these results, at least from the physical point of view.
Some short comments:
- Authors should comment on the meaning of "positive reduced effective cosmological constant Λ" (p. 5). This sentence does not mean anything for general readers without some clarification.
- Sentence on p. 5 "The only distinction between the various theories is that (2.9) admits multiple solutions that depending on the coupling constants αp." seems to be defective.
- The physical meaning of the solutions to (2.9) as corresponding to various distinct Lovelock vacua that are characterized by different effective cosmological constants does not seem to have a clear physical interpretation. The analysis of the Lovelock Lagrangian presented here is (hypothetically) framed in a QG framework but the cosmological constant (or constants, in fact) which appear in it are treated in a rather classical fashion. It is not clear what is the physical meaning of these solutions. Its connection with quantum vacuum is not explained beyond using the jargon of partition functions in Euclidean QG. It is therefore difficult to judge the physical significance of these "academic" calculations. What is the quantum gravity content of this vacuum framework? It is not clear the physical context for the phase transitions described in Sec. 5 . To which `universe' do they refer? It all looks very much as a formal exercise. I am afraid the authors should provide some physical context for this purely mathematical calculation. I could apply the same comments to Ref. [1], of course. The Euclidean approach to QG used in both cases is well known to be problematic and does not endow the cosmological constant with any particular status beyond the one that it already has in the LCDM model, where it is a mere parameter of the theory, with no obvious relation to quantum physics. In this sense, this approach is not particularly illuminating.
-Ref. [6] is not taken very seriously, despite the title. There is no information on it, apart from the name of the author.
- Authors should define "GSSS shell" as they do with other acronyms.
I have no special comment to make on the quality of English.
Author Response
Thank you for the report on our paper. We note that one referee is in favour of publication our paper as is, whereas the other referee had some suggestions for revision. We respond to the specific comments as follows.
1) Authors should comment on the meaning of "positive reduced effective cosmological constant Λ" (p. 5). This sentence does not mean anything for general readers without some clarification.
We have clarified the sentence on page 5 to address this issue.
2) Sentence on p. 5 "The only distinction between the various theories is that (2.9) admits multiple solutions that depending on the coupling constants αp." seems to be defective.
We have revised the sentence to read “The only distinction between the various theories is that (2.9) admits multiple solutions that depend on the coupling constants αp”
3) Ref. [6] is not taken very seriously, despite the title. There is no information on it, apart from the name of the author.
This has become a standard reference in the literature, acknowledging the idea presented in W. Fishler’s seminar proposing the finiteness of the dimension of Hilbert space in a de Sitter vacuum. We have therefore modified it only slightly to clarify the location of the seminar.
4) Authors should define "GSSS shell" as they do with other acronyms.
We have removed this acronym and have simply referred to the equation.
5) The only other concern referee #1 raised concerned the physical interpretation of the different vacuum solutions. We have modified the concluding section of our paper to address this point. Classically, only one of the possible de Sitter solutions will be physically realized — which one is arbitrary. However in the context of quantum gravity, the solution of maximal entropy will be realized until the volume of space expands to a point at which a different vacuum can have lower free energy (or more entropy), at which point there will be a phase transition to this other vacuum, as shown in figure 1.
Reviewer 2 Report
Comments and Suggestions for Authors
The authors discuss the partition function of higher-dimensional cosmological solutions of Einstein-Lovelock theory and the related thermodynamics. The paper is interesting and the results look correct, so I recommend the publication of this paper.
Comments on the Quality of English LanguageThe paper is well written, but contains several typos, like for example vaccua instead of vacua. A moderate revision is suggested.
Author Response
Thank you for the report on our paper. We note that this referee is in favour of publication our paper as is. We are grateful for the referee's acknowledgment of the importance of our work.